# The Oxygen Vacancy Defect of ZnO/NiO Nanomaterials Improves Photocatalytic Performance and Ammonia Sensing Performance

**DOI:** 10.3390/nano12030433

**Published:** 2022-01-27

**Authors:** Jiaqi Zhang, Jin Li

**Affiliations:** 1Xinjiang Key Laboratory of Solid State Physics and Devices, Xinjiang University, Urumqi 830017, China; jiaqiyuxiahua@stu.xju.edu.cn; 2School of Physical Science and Technology, Xinjiang University, Urumqi 830017, China

**Keywords:** ZnO/NiO, oxygen vacancy, photocatalysis, gas sensitivity, heterojunction

## Abstract

In this paper, ZnO/NiO composites rich in oxygen vacancies are prepared by the solvothermal method and reduction method. In the test, through the use of X-ray diffraction (XRD), X-ray photoelectron spectroscopy (XPS), scanning electron microscopy (SEM), transmission electron microscope (TEM), diffuse reflectance spectroscopy (DRS), photoluminescence spectroscopy (PL), and electron paramagnetic resonance (EPR), we effectively prove the existence of phase, morphology and oxygen vacancies in the material. Through the photocatalysis test and gas sensitivity test, it is found that 10% Ni doped OZN-10 has the best photocatalytic activity and gas sensitivity characteristics. The degradation rate of methylene blue (MB) was 98%. The gas sensitivity test shows that OZN-10 has good selectivity, good response performance (3000 ppm, 27,887%) and excellent response recovery time (response time: 50 s, recovery time: 5–7 s) for saturated NH_3_ gas at standard atmospheric pressure (101.325 KPa) and room temperature (25 °C). The synergistic effect of oxygen vacancy as the center of a trap and p–n heterojunction forming an electric potential field at the interface is explained, and the mechanism of improving photocatalysis and gas sensitivity is analyzed. This work will provide an innovative vision for dual-performance oxygen vacancy modification of heterojunctions through photocatalysis.

## 1. Introduction

Nanoparticles have attracted much attention because of their versatility, which can be used for photoelectric sensing, photocatalysis and gas sensor. Due to their large surface-to-volume ratio [1,2,3,4], they are widely prepared. Mixed metal oxide nanoparticles are composed of two or more metal oxide nanoparticles with enhanced unique properties. Dyes are widely used as colorants in various industries, such as textiles, leather, paper pulp, cosmetics, food and plastics, which can cause some environmental problems, especially water pollution. The water-soluble dye reduces dissolved oxygen levels and prevents light penetration, which can affect aquatic life and have serious ecological consequences. To this end, chemistry helps to understand the mechanisms of water pollution and to design procedures to overcome these problems without wasting natural resources. Turning contaminated water into fresh, healthy water is one of the best and low-cost ways to do this. Sunlight is one of the common elements in our environment [5].

ZnO is an N-type semiconductor with a band gap of 3.37 eV. Therefore, it acts as a photocatalyst under UV irradiation [6,7,8,9]. It has been reported that the photocatalytic efficiency of ZnO nanoparticles is higher than that of NiO nanoparticles for the degradation of methylene blue under visible light irradiation [10]. However, the material monomer is not suitable for all conditions, and the most important techniques for improving the efficiency of ZnO can be solved by its own defects or recombination with other photocatalytic oxides. Yang et al. found that the excellent photocatalytic performance in the degradation of perfluorooctanoic acid was attributed to the Bi_5_O_2_I/ZnO n–n heterojunction formed by calcination, which expanded the light response to the visible region and improved the separation efficiency of electron–hole pairs [11]. Song et al. fixed the direct Z-type heterojunctions composed of Ag_2_O nanoparticles and ZnO nanorods on the nickel foam (AZN) by combining hydrothermal and precipitation methods, successfully constructed three-dimensional mesh composites, and evaluated their photocatalytic performance under simulated sunlight. A series of photochemical test results show that the charge separation efficiency of AZN is significantly improved, which is attributed to the synergistic effect of direct Z-type heterojunction, matched band structure and three-dimensional porous structure [12].

Among the various defects observed in ZnO, oxygen (V_O_) and zinc (V_Zn_) vacancies deserve special attention for a variety of reasons. First, they are considered to be the defects with the lowest formation energy in ZnO, so their presence in ZnO is likely and observable [13]. Secondly, through experimental studies and theoretical methods, it is generally believed that these defects have one of the most prominent and critical effects on ZnO modification [14,15,16,17]. For example, the presence of oxygen vacancies has been reported to improve the availability of ultraviolet light, in addition to extending absorption edges into the visible spectrum, thereby improving the efficiency of various photocatalytic and photoelectrochemical processes, such as gas sensors, decomposition of organic pollutants and photocatalytic hydrogen production. Zhu et al. found that the excellent properties can be attributed to the contribution of oxygen vacancies in the two-step calcination samples to their photocatalytic properties. Electrons confined by oxygen vacancies can be transferred to O^−2^ to produce superoxide radicals, which can oxidize NO to the final product nitrate [18]. Zhang et al. found that the abundant oxygen vacancy, high specific surface area, and interfacial interaction between CeO_2_ and ZnO effectively amplified the resistance changes caused by oxygen adsorption changes. Two-dimensional (2D) porous structures also play an important role in excellent sensing performance due to excellent electron transport capacity and abundant gas diffusion channels [19].

Metal oxide p–n heterojunctions are widely used in photocatalysis and gas sensing, and many researchers have made very exciting progress. Umesh T. Nakate et al. found that electron and hole migration in p–CuO/n–TiO_2_ heterojunction devices occurred at the CuO/TiO_2_ interface, which generated a depletion zone leading to a potential barrier. The CO molecule on the sensor surface releases electrons into the conduction band of the sensor material. The released electrons recombine with the hole and transfer across the interface, forming a larger depletion zone width, leading to a higher barrier, and thus the resistance of the material continues to increase [20]. For zinc oxide, nickel oxide is a common semiconductor material with which a p–n heterojunction can be formed. Nickel oxide (NiO) nanoparticles exhibit photocatalytic activity because it is a p-type semiconductor with a wide band gap (3.6–4.0 eV) between the valence band and the conduction band. The degradation of organic dyes, such as methylene blue and rhodamine B, by NiO nanoparticles has been reported. Additionally, it has good gas-sensitive performance [21,22,23,24]. Bai et al. successfully synthesized rGO–NiO/ZnO composite materials by the hydrothermal method and hydrazine hydrate reduction method, and used them for the detection of NO_2_. Its excellent corresponding results are attributed to the formation of a p–n heterojunction, which benefits from the formation of an additional depletion layer at the junction interface, and the rGO decorative increase in surface-active sites and accelerate electron transfer [25]. The photocatalytic degradation of aniline blue (AB), bright green (BG) and direct red 80 (DR80) dyes by transition metal oxide nanocomposites with p–p isotype heterojunctions NiO/CuO and p–n heterojunctions CuO/ZnO and ZnO/NiO has been explored under sunlight and ultraviolet light [26]. Umesh T. Nakate et al. found that a p–n heterojunction established by p-type NiO nanosheets and n-type WO_3_ nanorods semiconductor nanostructures can effectively improve the sensitivity of the material to acetaldehyde, which shows a 2184% response to 100 ppm acetaldehyde at 250 °C, and shows a high degree of selectivity. It was also found that the decrease in the number of holes in NiO and the increase in the depletion width at the NiO/WO_3_ interface resulted in an increase in resistance [27].

In this work, we successfully prepare ZnO/NiO composites rich in oxygen vacancies by the solvothermal method and reduction method. The novel ZnO/NiO structure as photocatalytic and gas sensitive material is studied. The photocatalytic and gas sensitive results show that NiO and oxygen vacancy modified ZnO improve the photocatalytic and gas sensitive performance of the materials compared with pure ZnO. The presence of oxygen vacancies and p–n heterojunctions are thought to be responsible for the improved performance.

## 2. Materials and Methods

### 2.1. Materials

Ethylene glycol (EG), zinc acetate dihydrate (Zn(Ac)_2_·2H_2_O), nickel nitrate hexahydrate (Ni(NO_3_)_2_·6H_2_O), caustic potash (KOH), ethanol, sodium borohydride, tert-butanol, isopropanol, and potassium iodide were obtained from the Macklin company (analytically pure, 99%, Shanghai, China).

### 2.2. Synthesis of Spherical ZnO/NiO Nanoparticles with Oxygen Vacancy

The synthesis of nanosized oxygen vacancy zinc oxide/nickel oxide spherical particles is based on a previously reported modification strategy with ulterior motives. The specific synthesis process is shown in Figure 1, which is mainly divided into two steps. First, to prepare zinc oxide/nickel oxide nanospheres, 4 g of zinc acetate dihydrate and nickel nitrate hexahydrate (5, 10, 15%) in different mass percent were added to 50 mL of ethylene glycol in sequence. Then, it was stirred with a magnetic stirrer for 1 h until all was dissolved into a slightly green solution, and then the solution was transferred to a 100 mL stainless steel autoclave (Xi’an, China) and heated in a drying oven at 160 °C for 6 h. Afterwards, the obtained precursor was centrifuged and washed with alcohol and deionized water for 3 times, and then dried at 60 °C for 12 h, and the obtained green powder was denoted as ZN-5, ZN-10, ZN-15. In the second step, to create oxygen vacancies, three precursors with different contents and sodium borohydride were ground and mixed continuously for 1 h in a ratio of 2:1, and then placed in a CVD tubular furnace (OTF-1200X, Shenzhen, China). Then, the precursors were heated in N_2_ at a heating rate of 5 °C/min at 400 °C for 70 min. Then, the calcined powder was added to 1000 mL of deionized water. It was stirred well to dissolve the unreacted sodium borohydride and by-products. Finally, the precipitate was centrifuged, washed, and dried at 60 °C under vacuum to obtain OZN-5, OZN-10, and OZN-15. Afterwards, pure ZnO samples were prepared under the same conditions.

### 2.3. Characterization Instrument

The morphological characteristics of the object were observed by scanning electron microscope (SEM, LEO1430VP, Oberkochen, Germany); a photoluminescence (PL) spectrometer (PL, Hitachi F-4600, Tokyo, Japan) was used to judge the electron directional movement and electron-hole recombination; a transmission electron microscope (TEM, Tecnai G2 F20, Hillsboro, OR, USA) was used to determine the sample morphology, sample phase and sample 7 diffraction pattern; X-ray diffractometry (XRD, Bruker D8 Advance, Ettlingen, Germany) was used for the analysis of the sample phase; a UV-visible spectrophotometer (UV-Vis, Hitachi U-3010, Tokyo, Japan) was used for liquid and solid UV analysis; and X-ray photoelectron spectroscopy (XPS, Thermo ESCALAB 250XI, Newington, NH, USA) was used for elemental and valence band analysis. EPR strength was measured using an electron paramagnetic resonance spectrometer (EPR, EMXnano, Bruker, Ettlingen, Germany). Electrochemical impedance spectroscopy (EIS) was measured by ZAHNER-Im6ex (Kronach, Germany) electrochemical workstation. The workstation adopts a three-electrode mode with 1m KOH solution as electrolyte, Ag/AgCl as reference electrode, sample as working electrode and platinum as counter electrode. Photocurrent was measured using a three-electrode system, electrochemical workstation (CHI660, Shanghai, China) and xenon lamp (CELL-HXF300, Beijing, China). The photocatalytic workstation is equipped with a 350 W xenon lamp (λ = 300 nm; 0.56 W/cm^2^) and was used as the light source to test the photocatalytic activity of the prepared photocatalyst to decompose MB in XPA-7 photochemical reactor. A photoelectric integrated test platform was used to test the gas sensitivity performance (CGS-MT, Beijing, China). As shown in Figure 2, pure ZnO, ZN-10, OZN-10 and OZN-15 were loaded from left to right. The material was tested with saturated ammonia, ethanol, acetone, PA and H_2_O_2_. Through the conversion of ppm and mass fraction, a certain amount of saturated liquid was injected into a 500 mL conical bottle and put into a drying oven for natural volatilization until complete, and then the bottle was removed. Both ends of the electrode piece were clamped with an electrode clamp, and the performance of the electrode piece at room temperature and standard atmospheric pressure was tested by the CGS-MT instrument, a rubber plug was used to maintain a high concentration of 3000 ppm in the conical bottle. The electrode was quickly inserted into the bottle for response recovery test and the change of the current resistance of the sample was observed.

## 3. Results

### 3.1. Crystal Structure and Morphology of ZnO/NiO Nanoparticles

Figure 3 shows the XRD spectra of ZN-5, ZN-10, ZN-15, OZN-5, OZN-10 and OZN-15 nanoparticles. Black is the data of ZnO. Red is the data of NiO. The diffraction peaks in the XRD patterns are attributed to cubic NiO (JCPDS No. 73-1523) and hexagonal wurtzite ZnO (JCPDS No. 79-0205). The formation of cubic NiO is confirmed by diffraction peaks at 37.227, 43.253 and 62.828°, corresponding to planes (111), (200) and (220), respectively. The diffraction peaks near 31.850, 34.551, 36.357, 56.749 and 63.091° are due to the wurtzite structure of ZnO corresponding to the crystal planes of (100), (002), (101), (110) and (103), respectively. These diffraction peaks are wide and large, indicating that the particle size of the material is small and the grains are fine. The Debye–Scherrer formula was used to calculate the size of the microcrystals [28]. The formula is as follows:(1)$$D=\frac{K\cdot \lambda }{\beta \cdot cos\theta }$$
where *K* is Scherrer constant, *D* is the average thickness of grain perpendicular to the crystal plane, *β* is the half-peak height width or integral width of the diffraction peak of the measured sample, *θ* is the Bragg angle, and *λ* is the X-ray wavelength. It was found that the average calculated microcrystalline size of Zn-10 is 32–59 nm. The average value of OZN-10 is 18–37 nm. No impurity peaks were observed in the XRD spectra, confirming the composition of NiO and ZnO in all heterostructures.

The SEM images of the samples are shown in Figure 4. Figure 4a–e show the pure ZnO, ZN-5, ZN-10, ZN-15 and OZN-10, respectively. The agglomeration of particles is caused by the small size of the prepared particles, which is due to the fact that the synthesized particles have no specific morphology. In addition, the high surface energy of particles in the synthesis process can also be seen in the sample, so particles tend to reduce the surface energy through agglomeration. As can be seen from Figure 4a, ZnO is composed of many stacked nanorod-like particles and the particles are small. It is observed in Figure 4b–d that, with the increase in nickel element, the particles change from a rod shape to spherical shape, and the higher the amounts of the nickel element, the particles become rounder, so it can be inferred that the particles change from a rod shape to spherical shape. The addition of the nickel element acts as an element that inhibits the growth of axis C, so that axis A, B and C become uniform growth. It was found that, with the increase in the nickel content, particle agglomeration decreases and particle size distribution is relatively uniform. Figure 4e shows nano OZN-10 particles rich in oxygen vacancies after calcination and reduction by NaBH_4_. The reduced OZN-10 particles maintained their original structure, but their diameters were reduced, indicating that NaBH_4_ treatment did not destroy the ZnO particle structure, but only reduced the particle diameters by calcination. The diameter is reduced from about 60 nm to about 40 nm, and the calcination temperature is relatively high, which makes the crystal crystallinity better and thus the grain size decreases. After reduction, the specific surface area of samples is increased, the contact between effective substances and pollutants and gases is enlarged, and the photocatalytic performance and gas sensitivity are improved. The corresponding EDS spectrum is shown in Figure 4f. The peaks of other elements, except Zn, Ni and O, are strong, and the peaks of other elements are slightly lower. The signal peak of element C comes from the conductive adhesive during the test, which further confirms the high purity of the prepared NiO/ZnO composite nanoparticles. In addition, the atomic ratio of nickel to zinc detected was 9.39%, slightly lower than that of the initial solution (10%). This may be due to loss of nickel during the reaction and washing. The oxygen content is also lower than normal, indicating the loss of oxygen and the existence of a large number of oxygen vacancies.

Figure 5 shows TEM images, TEM mapping and selected electron diffraction images of the OZN-10 catalyst, respectively. As shown in the figure, the sample is also composed of many spheroid nanocomposites, forming a hierarchical porous structure with an average size of 40 nm. These nanoparticles are loosely clustered together to form many irregular gaps, which is consistent with the appearance of ZnO/NiO nanoparticles, as shown by the SEM images, and is favorable for photocatalysis and gas sensitive reaction. TEM mapping further confirmed the co-existence of Zn (red), Ni (blue) and O (green) at the nanoscale. The figure shows that NiO and ZnO are well combined, and the nickel element is distributed evenly throughout the sample. The content of zinc element in the material is higher than that of nickel element, which is in line with the pre-defined 10% addition of nickel element. TEM shows an evident crystal structure and lattice stripes, among which 0.247 nm corresponds to ZnO in-plane (101) plane, and 0.209 nm corresponds to NiO in-plane (200) plane. By the TEM analysis of another particle, it can be found that the mid-plane spacing of 0.281 nm corresponds to the in-plane (100) of ZnO, while the mid-plane spacing of 0.241 nm corresponds to the in-plane (111) of NiO. Moreover, the lattice spacing of some ZnO at the interface of the material is increased to about 0.25 nm [29]. The results indicate that some Ni^2+^ may replace Zn^2+^ and disperse at the grain boundary between ZnO particles and NiO, forming sites or gaps in ZnO lattice. In addition, there are many vertical cross lattice fringes at the grain boundary in the figure, indicating that the interface between ZnO and NiO is very tight, and there are heterojunction structures between n-type ZnO and p-type NiO, which is beneficial to charge separation and improve the photocatalytic performance of the sample. The SAED diagram shows that ZnO/NiO nanostructures have a polycrystalline structure with circular diffraction points, corresponding to different crystal planes of ZnO and NiO, respectively. The scattering of white dots is also observed in the figure, and it is revealed that the prepared sample is again in the highly crystalline and wurtzite structure of the sample. In the figure, the 3 outer rings are mainly characteristic crystal planes of zinc oxide (004), (203) and (103), while the 2 inner rings are mainly characteristic crystal planes of nickel oxide (200) and (111). The above analysis confirmed the successful synthesis of nano-ZnO/NiO particle structure.

### 3.2. Valence Analysis of Chemical Composition and Elements

The chemical composition and elemental states of pure ZnO, ZN-10 and OZN-10 samples were measured by X-ray photoelectron spectroscopy (XPS). ZN-10 and OZN-10 samples showed the presence of Zn, O, Ni and C in the whole XPS spectrum, and no other elements were found. The presence of component C is mainly attributed to its absorption in the atmosphere before XPS measurement. Zn 2p, O 1s, and Ni 2p in the high-resolution XPS are shown in Figure 6. The two peaks of 1021.5 eV and 1044.6 eV in the Zn 2p spectrum due to the split between the spin orbits can be attributed to the double lines of Zn 2p_3/2_ and Zn 2p_1/2_, which are consistent with the state of Zn^2+^ in ZnO. We observed the strongest Zn 2p peak in pure ZnO, and the addition of nickel led to the weakening of the Zn 2p peak. It may be caused by the partial NiO covering on ZnO nanoparticles. In Figure 6b, compared with pure ZnO samples, the Zn 2p spectrum of the ZnO nanorods doped with NiO is shifted to a higher binding energy direction of about 0.4 eV. This change in binding energy is attributed to the strong interaction between ZnO and NiO in the ZnO/NiO heterostructure, thus demonstrating the formation of heterojunctions in the material. In the figure, the Ni 2p signals of ZN and OZN can also be deconvolved into five peaks. The peak of Ni 2p_3/2_ 852.87 eV indicates the presence of Ni^3+^. The remaining peaks indicate the presence of Ni^2+^, and the comparison between ZN and OZN shows that Ni^3+^ is also reduced to Ni^2+^ after the reduction in NaBH_4_. Compared with the Ni 2p peaks (2p_3/2_ and 2p_1/2_) of ZN nanoparticles, the Ni 2p peaks of OZN move towards the lower binding energy, which is about 0.6 eV. Traditionally, binding energy shifts in XPS spectra have been explained by two mechanisms: different electronegativity of metal ions and strong interactions between nanocrystals. The electronegativity of Ni^2+^ is about 1.91, which is about 0.26 than that of Zn^2+^. Ni^2+^ can absorb electrons from Zn^2+^, so the shielding effect of electrons on Zn^2+^ is reduced, but the shielding effect on Ni^2+^ is increased [30]. Thus, the Zn 2p peak moves in the direction of higher binding energy, while the Ni 2p peak moves in the direction of lower binding energy. This indicates that there may be Ni–O–Zn bond at the interface between NiO and ZnO nanoparticles, which further confirms the TEM results. In addition, when NiO and ZnO nanoparticles are bound together, electrons can be transferred from n-type ZnO to p-type NiO nanoparticles until the system reaches equilibrium. Therefore, the Zn 2p peaks and Ni 2p peaks in our experiment move towards higher and lower binding energies, respectively. The results show that there may be strong interactions between p-type NiO and n-type ZnO nanoparticles. These results indicate that p–n heterojunctions are formed in ZnO/NiO nanoparticles. In addition, as shown in the figure, the main peak at 530.3–530.6 eV can be allocated to the lattice oxygen of ZnO, and the signal at the higher binding energy of 532.0–532.4 eV can be attributed to the formation of oxygen vacancies on the surface [31]. As can be seen from the figure, defect states also appear on the surface of pure ZnO due to its small particle size, leading to the appearance of oxygen vacancies. With the addition of Ni, the content of oxygen vacancy increases obviously, which can be attributed to the presence of V_O_ or V_Zn_ in the ZnO lattice accompanied by the doping of metal ions. The free competitive growth pattern of Zn–O and Ni–O bonds at the Zn–O–Ni interface leads to a large number of oxygen vacancies, and the lattice perturbation and reconstruction are induced by doping. In addition, the ionic radii of zinc and nickel are different, and the addition of Ni to ZnO nanoparticles leads to the destruction of the local equilibrium of the crystal structure. This causes an increase in internal stress in the main lattice, and in order to release the stress, oxygen vacancies need to be formed. In OZN-10, the strongly reductive NaBH_4_ was calcined in an N_2_ atmosphere. Compared with the previous two figures, we successfully introduced a large number of oxygen vacancies in ZnO/NiO. These data are in good agreement with previous reports on ZnO. Oxygen vacancies are introduced to the surface at high temperatures, hypoxia and reductive materials.

### 3.3. Photophysical Property of Samples

Diffuse ultraviolet-visible spectroscopy is used to study the optical adsorption performance of ZnO samples, and the corresponding spectra are shown in Figure 7. As shown in the figure, due to its inherent structural characteristics, pure ZnO shows typical ultraviolet light absorption (λ < 390 nm). Interestingly, with the introduction of the nickel and oxygen vacancy, the absorption edges of ZN and OZN series samples show a significant red shift, which is attributed to the fact that the surface oxygen vacancy can create a defect isolation above the ZnO valence band. From the diffuse reflectance spectrum and the appearance of the sample, it is also consistent, because the addition of oxygen vacancies causes the color of the sample to darken, from light green to black. It can be seen from the spectrum that the absorbance of OZN series samples in 390–780 nm is significantly higher than that of ZN and Pure ZnO, which effectively improves the absorption efficiency of visible light. In addition, according to the Kubelka–Munk equation, the band gaps of pure ZnO, ZN-5, ZN-10, ZN-15, OZN-5, OZN-10 and OZN-15 composites can be estimated by the following formula, (*αhv*)2 = A(*hv* − E_g_), with which the bandgap energy (E_g_) is calculated, where *α*, *hv* and A are absorption coefficients, photon energy and constants. Therefore, the calculated band gaps of pure ZnO, ZN-5, ZN-10, ZN-15, OZN-5, OZN-10 and OZN-15 are 3.21, 3.18, 3.12, 3.17, 3.07, 2.88 and 2.92 eV, respectively. The order of band gap is consistent with the previous analysis; the band gap of OZN series is smaller than that of ZN series, and the introduction of oxygen vacancy is beneficial to enhance the light absorption [32]. The addition of oxygen vacancies causes electrons to jump from the valence band to the local state and from the isolated state to the conduction band, resulting in visible light absorption.

Photoluminescence spectroscopy is an effective method to study the electronic structure and optical properties of semiconductor nanomaterials. It can reveal the structural properties of materials, such as surface oxygen vacancy and defect, surface states, and separation and recombination of photogenerated carriers, thus providing a strong basis for the development and preparation of high-performance semiconductor functional materials. In Figure 8, the PL curves of pure ZnO, ZN-5, ZN-10, ZN-15, OZN-5, OZN-10 and OZN-15 samples at the excitation wavelength of 320 nm are summarized. The UV luminescence peak of ZnO is located at 390 nm, corresponding to the near band edge luminescence of ZnO with a band gap energy of 3.21 eV. This result corresponds to the characterization analysis of the band gap energy in DRS above. In addition, other samples except pure ZnO have a strong and wide visible luminous band at 450–500 nm, which can be considered to be generated by the transition of electrons from the excited state to the valence band from the oxygen defect level. Compared with pure ZnO, the fluorescence intensity of the NiO/ZnO composite decreases evidently, and the intrinsic emission peak of NiO/ZnO also widens. This indicates that the introduction of NiO significantly inhibits the recombination probability of photoelectron–hole pairs, and the effective charge separation can increase the lifetime of carriers and improve the efficiency of charge transfer to the material interface, thus improving the photocatalytic performance of photocatalyst. It was also found that there were oxygen defect levels on the surface of ZN and OZN series samples, mainly oxygen vacancy. ZnO samples had small particle size, short mean free path of electron motion, and oxygen vacancy defects on the ZnO surface. Thus, the defect is located on the surface of the particle [26]. After Ni^2+^ doping, the emission of ZN and OZN is greatly inhibited. It is possible to increase the electron transfer rate between ZnO nanoparticles and Ni^2+^, thereby improving the interface interaction between them. Thus, the Ni^2+^ and oxygen defects inhibit electron–hole pair recombination and accelerate electron transfer.

### 3.4. Analysis of the Photocatalytic Properties of ZN and OZN

Pure ZnO, ZN-5, ZN-10, ZN-15, OZN-5, OZN-10, and OZN-15 nanoparticles were used to degrade MB dye in order to evaluate the photocatalytic performance of the prepared ZN series and OZN series catalysts. We prepared 30 mg/L methylene blue solution and added 10 mg photocatalyst to it for reaction. The photodegradation rate was calculated by using the equation: D(%) = (1 − C/C_0_) × 100%, where C_0_ is the initial MB concentration and absorbance before illumination, respectively, and C is the MB concentration and absorbance at t (min) after illumination, respectively. Figure 9 shows the UV-Vis absorption spectra of the MB solution degradation process using pure ZnO, ZN-5, ZN-10, ZN-15, OZN-5, OZN-10 and OZN-15 as catalysts. As it can be seen from Figure 9, the characteristic absorption peak of MB is about 665 nm. With the extension of illumination time, the characteristic absorption peak of MB decreases significantly, and no new absorption peak is generated. It can be seen from the figure that OZN-10 has the best degradation efficiency, and the curve is close to a horizontal straight line at 100 min. Compared with pure ZnO, there is a great difference, and pure ZnO also has a large characteristic peak. The time dynamic degradation efficiency evolution of residual MB concentration in photocatalyst solution is shown in the figure. In accordance with the results of liquid UV, the decolorization efficiency of pure ZnO photocatalyst is about 46.82%. Due to the advantages of ZN and OZN, the smaller particle size provides a larger specific surface area and the presence of oxygen vacancies. The ZN samples show higher photocatalytic activity than pure ZnO samples. However, the degradation of MB by OZN is more significant, and OZN-10 samples show the best photocatalytic performance; the efficiency is about 98.05%. The surface degradation efficiency and rate constant K of various samples were calculated using the data in the figure, and the results were summarized in the table. The degradation efficiency of pure ZnO, ZN-5, ZN-10, ZN-15, OZN-5, OZN-10 and OZN-15 are 46.82, 80.16, 87.50, 80.57, 86.18, 98.05 and 91.98%, respectively. The degradation efficiency of OZN-10 is the highest among all the analyzed samples, and the amount of degradation in the same time is about twice that of pure ZnO. The photocatalytic reduction process was characterized by the quasi-first-order kinetic model equation: −ln(C/C_0_) = kt, where t is time (min). The rate constants k of pure ZnO, ZN-5, ZN-10, ZN-15, OZN-5, OZN-10 and OZN-15 are 0.22 × 10^−2^, 0.56 × 10^−2^, 0.67 × 10^−2^, 0.55 × 10^−2^, 0.65 × 10^−2^, 1.2 × 10^−2^, 0.8 × 10^−2^, respectively, and the reaction rate with OZN-10 was 1.2 × 10^−2^ min^−1^, which was about six times that of pure ZnO sample. The linear relationship confirmed that the photocatalytic behavior of MB was completely consistent with the quasi-first-order kinetics under these reaction conditions. This enhanced ability is attributed to the reduction in the recombination ability of electron hole pairs due to the potential plant of heterojunction generation. Additionally, it can be attributed to the narrow band gap extending the light response to the visible region. OZN has a small particle size in its structure, which can provide a large reaction space and numerous active sites. Because of the formation of surface oxygen vacancy defects, it has a good electron hole separation efficiency and long excitation life.

### 3.5. Photoelectric Chemical Performance Test and EPR Analysis

EPR can qualitatively and quantitatively detect unpaired electrons contained in atoms or molecules of matter and explore the structural characteristics of their surrounding environment. For free radicals, orbital magnetic moments have almost no effect, and most of the total magnetic moments come from the contribution of electron spin. EPR can accurately confirm the existence and status of oxygen vacancies in materials. Because EPR can detect unpaired electrons and bound single-electron oxygen vacancies, the corresponding g factor of bound single-electron oxygen vacancies is generally between 2.001 and 2.004, and the corresponding g factor value of bound single-electron oxygen vacancies is generally considered to be in the bulk phase rather than the surface. The surface oxygen vacancies can be analyzed in combination with the previous XPS, and the nanostructures of pure ZnO, ZN-10 and OZN-10 were tested to further study the inherent defect centers generated during the modification of ZnO nanostructures. Figure 10 shows that pure ZnO, ZN-10 and OZN-10 have a paramagnetic defect center, and ZnO has an EPR vibration curve at g = 2.003. It was previously reported that g = 2.001–2.004 was attributed to surface oxygen vacancy, indicating the existence of oxygen vacancy defects in the sample. As shown in the figure, the EPR spectra of pure ZnO, ZN-10 and OZN-10 show that the resonance peak intensity of OZN-10 and ZN-10 is higher than that of pure ZnO, and the vibration peak intensity of pure ZnO < ZN-10 < OZN-10, indicating that OZN-10 calcined by NaBH_4_ has the most oxygen vacancy defects. The more oxygen vacancy defects, the smaller the band gap [33]. This observation is consistent with UV-Vis data because the oxygen vacancy defect enhances visible light absorption and thus reduces the band gap.

Photochemical measurements of ZnO, ZN-10 and OZN-10 were used to evaluate the electron-hole separation efficiency. Oxygen vacancy enhances visible light absorption and thus reduces the band gap. Figure 11 shows the variation of photocurrent over time for different samples at zero bias. The results show that the photocurrent of ZN-10 and OZN-10 samples is significantly stronger than that of ZnO under visible light irradiation, and the photocurrent is further enhanced with the addition of oxygen vacancy. This can be attributed to the introduction of oxygen vacancy defect levels, and the addition of effectively shortened band gap NiO to form a heterojunction, which greatly reduces the recombination of photocarriers and enhances the absorption in the visible light range. As it can be seen from the figure, the arc radius of the EIS Nyquist diagram of the OZN-10 composite is smaller than that of ZN-10 and pure ZnO, indicating that the OZN-10 composite has a higher photogenerated carrier separation efficiency and synergistic effect of oxygen vacancy and heterojunction generated by NiO. It is known that a high electron hole separation efficiency leads to a higher photocatalytic activity, which means that NiO composites have a stronger photocatalytic performance than other samples [34].

### 3.6. Chemical Potential Energy and Free Radical Experiments

According to the tangent line of data line obtained in Figure 12, it can be concluded that the valence band potential of pure ZnO, ZN-10 and OZN-10 is 2.75, 2.70 and 2.65 eV, respectively. This indicates that, compared with ZnO, the valence band potential of modified OZN-10 moves to a negative potential, which can reduce the band gap width and the adjusted valence band potential, and enhance the oxidation of photogenerated holes. In addition, according to the UV-Vis diffuse reflectance and absorption spectra, for pure ZnO, ZN-10 and OZN-10, the formula E_cb_ = E_vb_ − E_g_ can be calculated as −0.46 eV, −0.42 eV and −0.23 eV, respectively. According to the reaction potentials H_2_O/•OH (2.28 eV NHE) and OH^−^/•OH (1.99 eV NHE), it can be concluded that the valence band potential of OZN-10 can cause H_2_O or OH^−^ to produce •OH. On the other hand, the valence band conduction potential of OZN-10 is −0.23 eV, which is positive than O_2_/•O^2−^ (−0.33 eV NHE), so •O^2−^ cannot be formed, but is negative than O_2_/H_2_O_2_ (+0.69 eV NHE), so the reaction can be carried out. Under the excitation of visible light, H_2_O or OH^−^ in the liquid produces •OH, and O_2_ reacts to generate H_2_O_2_ [20]. Such double oxidation effectively improves the reaction efficiency of photocatalysis. In order to further verify the photocatalytic mechanism of OZN-10, the main reaction species in MB photodegradation system were detected by free radical capture experiment. As shown in Figure 13, •OH, •O^2−^ and h+ were captured by isopropanol (IPA), p-benzoquinone (BQ) and potassium iodide (KI), respectively. After the addition of scavengers in the photocatalytic reaction system, the photodegradation rate of MB of KI and IPA decreased most evidently, and the photocatalytic activity of OZN-10 was inhibited. As it can be seen from the figure, the photocatalytic degradation efficiency of MB decreased by about 90% when holes and captured, indicating that holes and •OH are the main free radical substances in OZN-10 system, which matches the chemical potential analysis.

### 3.7. Photocatalytic Mechanism of OZN Materials

Based on the above discussion and previous studies, the degradation mechanism of MB by OZN-10 photocatalyst is shown in Figure 14. When simulated sunlight illuminates the photocatalyst, valence band (VB) electrons on OZN-10 are excited rapidly and transferred to the conduction band (CB) based on a reduced band gap (2.88 eV). Traditionally, photogenerated electrons and holes can recombine quickly. However, due to the presence of oxygen vacancies, photogenerated electrons can be effectively captured and combined with the oxygen adsorbed on the surface of OZN-10 to form •O^2−^, which is very helpful in the depolymerization of intermediates into valuable products due to its moderate redox potential. On the other hand, because photoelectrons are captured effectively, electrons and holes are separated effectively. At the same time, the photogenic hole reacts with OH^−^ in the solution to form •OH. Because •OH exhibits a high oxidation potential, it cannot selectively oxidize MB to intermediates and further oxidize to CO_2_ and H_2_O. In addition, photogenic holes can also oxidize MB directly into intermediates and further oxidize CO_2_. On the other hand, the introduction of oxygen vacancies not only improves the separation efficiency of optical carriers as trap centers, but also produces defect-related states near the valence band, resulting in a narrower band gap and enhanced light absorption. In fact, the doping of Ni^2+^ and the appropriate oxygen vacancy concentration promote the production of •O^2−^ to a certain extent [35]. It is generally believed that the increase in oxygen vacancy concentration is beneficial to the capture of photogenerated electrons, thus improving the separation ability of photogenerated electrons and holes, and promoting the photocatalytic degradation of dyes and antibiotics into CO_2_ and H_2_O [33]. It is also known that the photocatalytic activity of the photocatalyst depends mainly on whether the electron–hole pair can be effectively separated. On the surface of a photocatalyst, photoexcited electrons and holes can change in various ways. Among them, capture and reorganization are the most important competition processes. Photocatalytic reactions are effective only when the photoexcited electron–hole is captured. Without a suitable trapping agent for electrons or holes, they recombine with each other and release heat inside or on the surface of the semiconductor. In order to improve the photocatalytic activity of ZnO, two important approaches need to be considered according to the separation mechanism of electrons and holes. It is well known that ZnO is an n-type semiconductor and when NiO is doped into ZnO particles, many tiny p–n junctions are formed in the photocatalyst NiO/ZnO. At equilibrium, the internal electric field formed makes the p-type NiO region negatively charged and the n-type ZnO region positively charged. Electron–hole pairs may be created under near-ultraviolet light. Under the action of the internal electric field, the hole flows to the negative field, and the electrons move to the positive field. Thus, photoelectron–hole pairs will be effectively separated by p–n junctions formed in NiO/ZnO. In addition, according to the position of the band edge, the photoexcited holes in the valence band of NiO are transferred to the valence band of NiO because the valence band of NiO is lower than that of ZnO, and the photoexcited electrons of ZnO remain in the conduction band of ZnO. Photoelectron–hole pairs will be effectively separated. Therefore, the enhanced photocatalytic performance of the p–n junction photocatalysts is attributed to the internal electric field assisted charge transfer at the junction interface between semiconductors with matching potential, thus facilitating the effective photoexcited electron–hole separation between the two semiconductors. Meanwhile, p-type NiO species have been reported to act as cavity traps and collectors. Therefore, there are enriched electrons on the surface reacting with MB adsorbed on the surface of the photocatalyst.
(2)$$\mathrm{NiO-ZnO}+\mathrm{hv}\;\to \;\mathrm{NiO-ZnO}(e_{CB}^{-}+h_{VB}^{+})$$
H_2_O → H^+^ + •OH(3)
(4)$$\mathrm{NiO}(e_{CB}^{-})\;\to \;\mathrm{ZnO}(e_{CB}^{-})$$
(5)$$\mathrm{NiO}(h_{VB}^{+})\;\to \;\mathrm{ZnO}(h_{VB}^{+})$$
(6)$$\mathrm{NiO-ZnO}(e_{CB}^{-})+H^{+}+O_{2}\;\to \;H_{2}O_{2}$$
(7)$$H_{2}O_{2}+e^{-}+\mathrm{hv}\;\to \;•\mathrm{OH}+{\mathrm{OH}}^{-}$$

### 3.8. Analysis of Gas Sensitive Characteristics

In order to prove that adding NiO and oxygen vacancy into ZnO nanoparticles is an effective method to enhance the gas sensitivity of ZnO-based gas sensors, the gas sensitivity of pure ZnO, ZN-10, OZN-10 and OZN-15 sensors were tested. Surveys were conducted separately. The sensor response to saturated NH_3_ gas was accurately measured at room temperature and standard atmospheric pressure to explore the role of oxygen vacancy and NiO doping, as shown in Figure 15. The effective material loading of the sensor material is about 3 mg. The change in gas response versus operating time was clearly observed for all sensors. Pure ZnO, ZN-10, OZN-10 and OZN-15 responded to saturated ammonia gas. It was found that the introduction of NiO and the addition of oxygen vacancy effectively improved the response of the sample to ammonia gas. After three reversible cycles, the sensor response of the four sensors showed excellent repeatability. For pure ZnO, the performance of OZN-15 improved by about 90%. Anti-interference is an important characteristic of sensor. Therefore, common volatile compounds, such as alcohol, acetone and formaldehyde H_2_O_2_, were tested to explore the selectivity of the sensor. As shown in Figure 16, the response of the sensors to saturated solutions except NH_3_ is low, which is 0.2% of the response of saturated NH_3_. This can be attributed to the similar characteristics of the electrons of NH_3_. The other mentioned systems can be ignored and hardly affect the detection of NH_3_. OZN-10 showed good selectivity for ammonia gas. V_O_ acts as the preferred adsorption site for NH_3_ molecules. It causes the extraction or capture of electrons from the conduction band, resulting in a change in resistivity. The more oxygen vacancies are produced, the more sensitive the gas is, because a higher concentration of NH_3_ molecules can be adsorbed. Oxygen vacancies provide additional electrons to participate in the response and recovery process, thus improving the adsorption of ZnO to gas molecules. Moreover, the formation of a p–n heterojunction at the interface between n-type ZnO and p-type NiO plays an important role in improving the performance. Due to the difference in band gap and Fermi level between the p-type NiO and n-type ZnO, they form a p–n heterojunction at the interface through charge carrier transport, thus increasing the barrier height and generating an additional depletion region [35]. As a result, the p–n heterojunction formed when the composite is exposed to air enables the sensor to exhibit a higher resistance. However, when ammonia is introduced, more trapped electrons are released back into the composite through the interaction between the adsorbed oxygen ions and the reducing gas molecules, resulting in a significant contraction of the depletion zone of the p–n junction and a decrease in the barrier height. As a result, the sensor resistance drops sharply, resulting in a significant change in resistance and an enhanced ammonia response.

## 4. Discussion

In conclusion, ZnO/NiO nanoparticles containing oxygen vacancy defects were prepared by the solvothermal reduction and high temperature reduction. The small particle size and large specific surface area of nanoparticles can greatly improve their optical, photocatalytic and gas sensitive properties. OZN-10 showed excellent photocatalytic performance driven by solar energy and almost completely degraded methylene blue. The degradation efficiency of OZN-10 is the highest among all the analyzed samples, and the amount of degradation within the same time is about twice of that of pure ZnO, which is related to the heterojunction structure and surface oxygen vacancy defects of OZN-10. In addition, the good response and selectivity of the material to NH_3_ were also found through gas sensitivity tests. The adopted synthesis strategy opens up interesting perspectives for the design of a p–n heterojunction and oxygen vacancy composite nanostructures for different types of applications, from selective sensor devices to high-performance photocatalysts.

## Figures and Tables

**Figure 1 nanomaterials-12-00433-f001:**
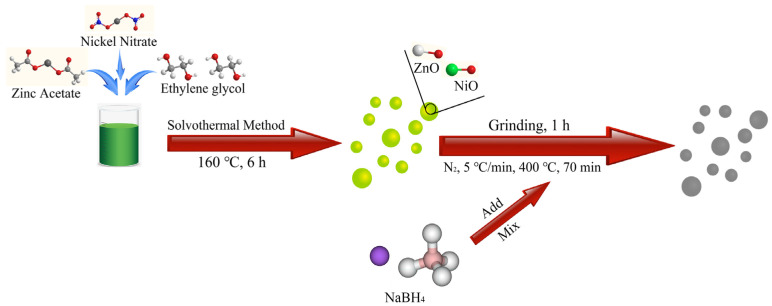
Schematic synthesis of ZnO/NiO nanoparticles with stable oxygen vacancy structures.

**Figure 2 nanomaterials-12-00433-f002:**
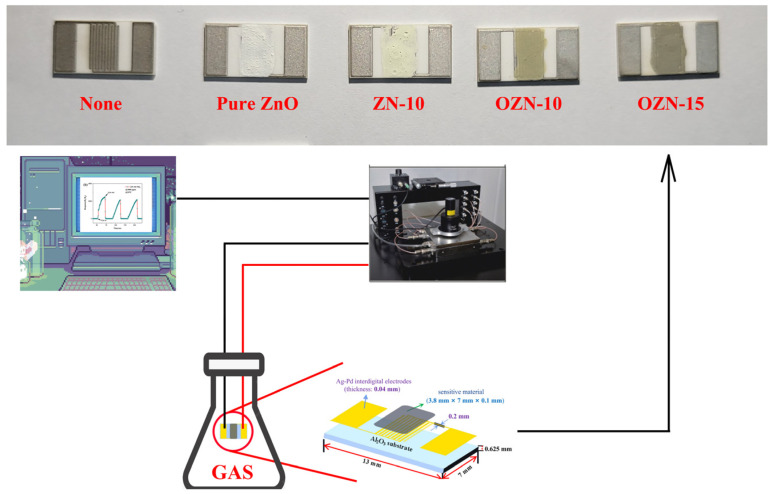
Schematic of the alumina substrate, sensor devices and gas sensor testing experimental system.

**Figure 3 nanomaterials-12-00433-f003:**
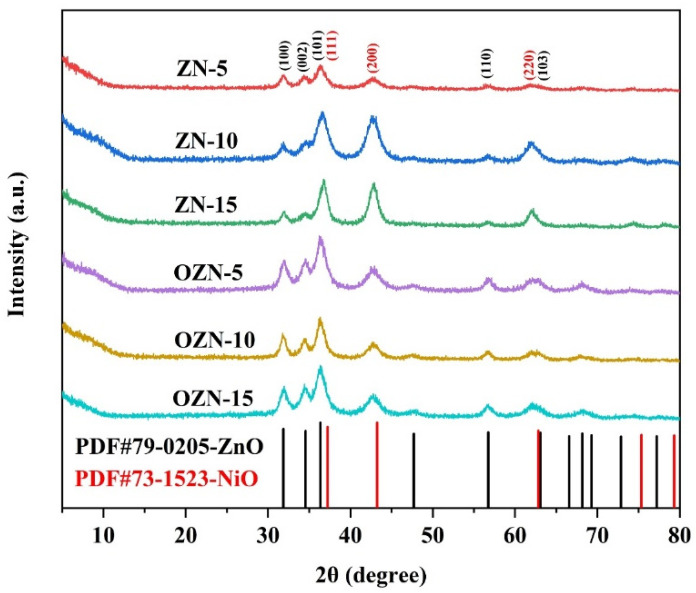
XRD patterns of different Ni content and oxygen vacancy addition.

**Figure 4 nanomaterials-12-00433-f004:**
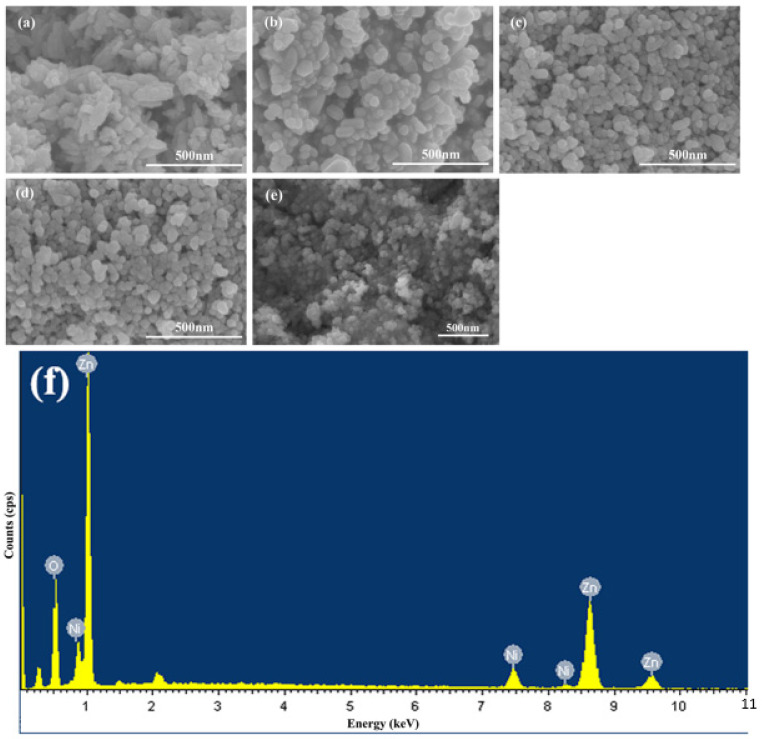
SEM images of (**a**) ZnO, (**b**) ZN-5, (**c**) ZN-10, (**d**) ZN-15, (**e**) OZN-10, and (**f**) EDS spectra of OZN-10.

**Figure 5 nanomaterials-12-00433-f005:**
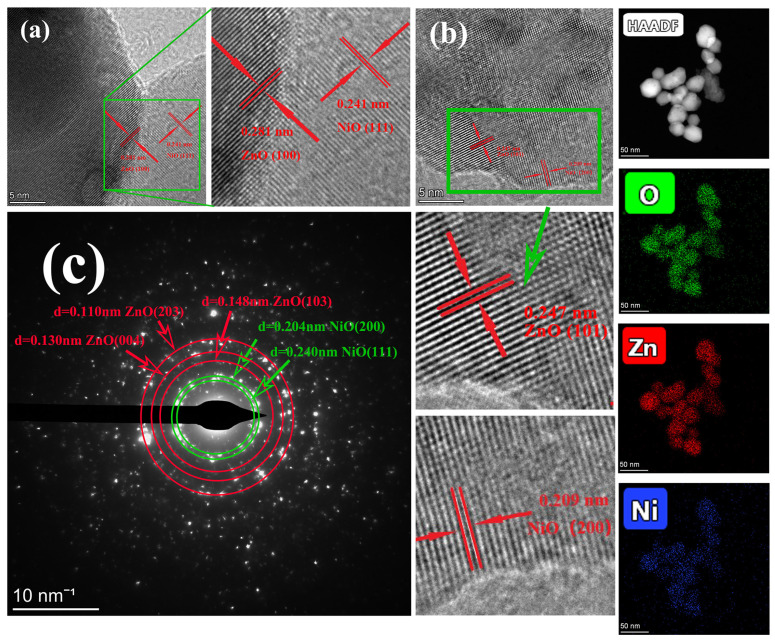
HRTEM images of (**a**,**b**) OZN-10; (**c**) SAED images of OZN-10; and TEM-mapping image of Zn (red), Ni (blue) and O (green).

**Figure 6 nanomaterials-12-00433-f006:**
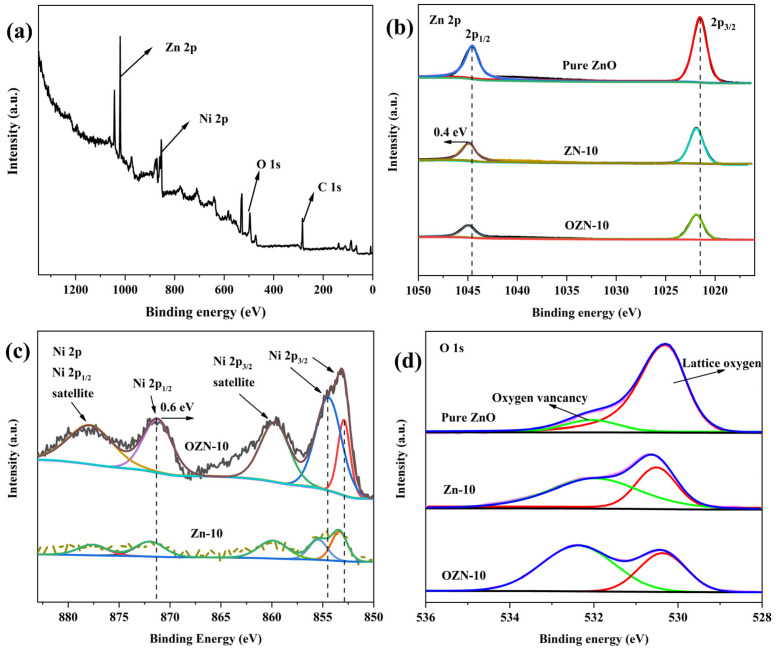
XPS spectra of OZN-10 (**a**) full-range scan of the samples, (**b**) Zn 2p core level, (**c**) Ni 2p core level and (**d**) O 1s core level.

**Figure 7 nanomaterials-12-00433-f007:**
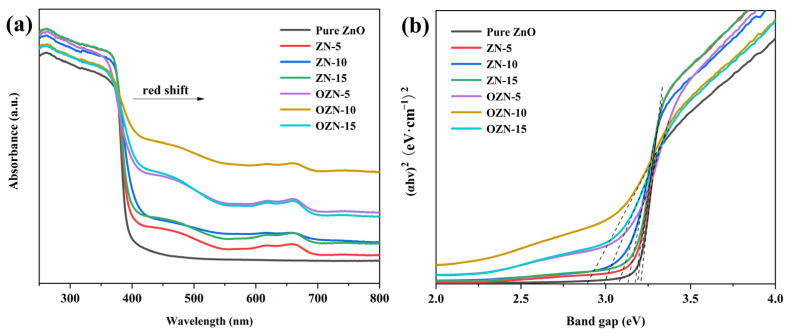
(**a**) UV-Vis absorption spectra and (**b**) band gap energy of ZnO, ZN and OZN.

**Figure 8 nanomaterials-12-00433-f008:**
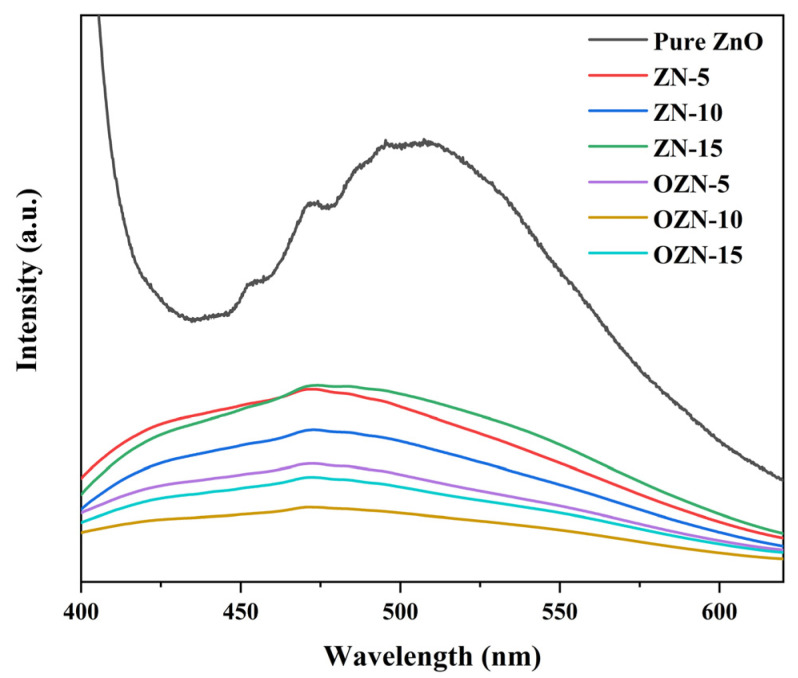
PL spectra of ZnO, ZN and OZN nanoparticles.

**Figure 9 nanomaterials-12-00433-f009:**
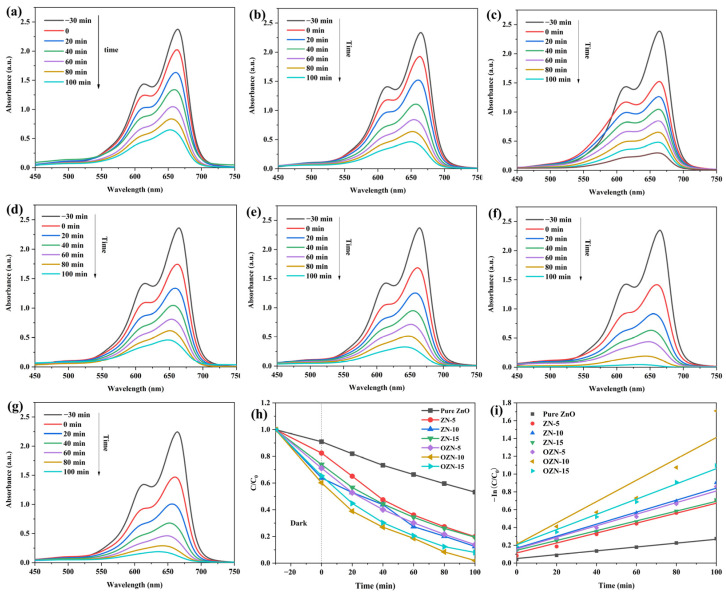
In the presence of 10 mg (**a**) ZnO, (**b**) ZN-5, (**c**) ZN-10, (**d**) ZN-15, (**e**) OZN-5, (**f**) OZN-10, and (**g**) OZN-15, the absorption spectra of 30 mg/L methylene blue solution under visible light irradiation. (**h**) Degradation efficiency of pure ZnO, ZN and OZN samples. (**i**) First-order kinetics of degradation of pure ZnO, ZN and OZN samples.

**Figure 10 nanomaterials-12-00433-f010:**
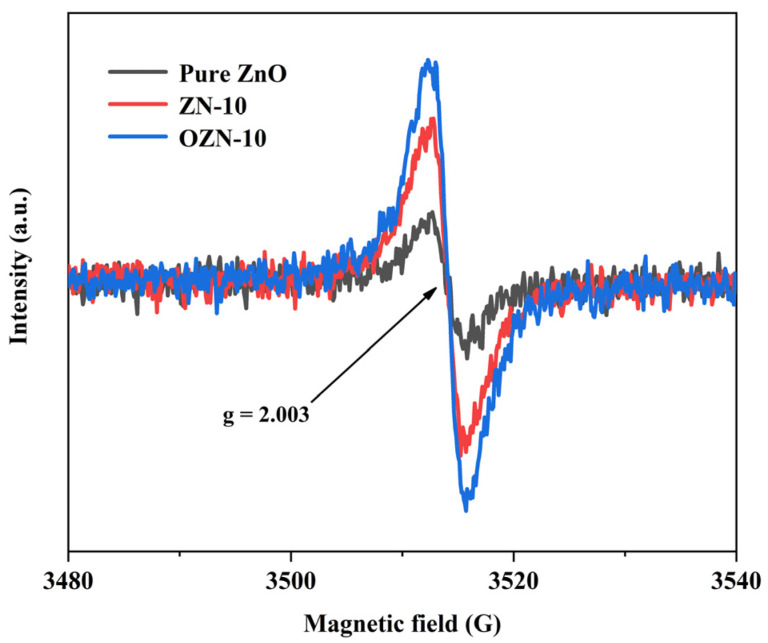
EPR spectra of pure ZnO, ZN-10, OZN-10.

**Figure 11 nanomaterials-12-00433-f011:**
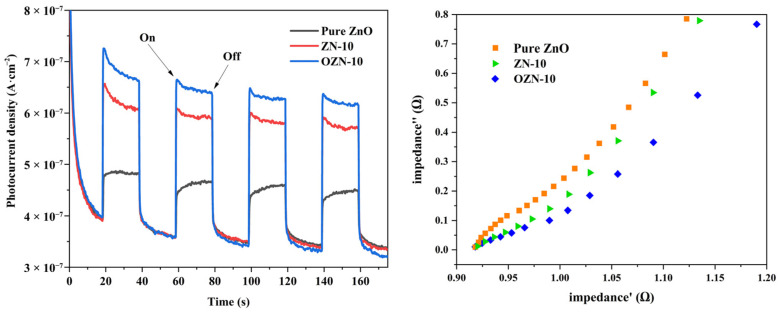
Photoelectrochemical properties of pure ZnO, ZN-10 and OZN-10 and (**a**) photoresponse diagram of time amperometry. (**b**) Nyquist plot of electrochemical impedance.

**Figure 12 nanomaterials-12-00433-f012:**
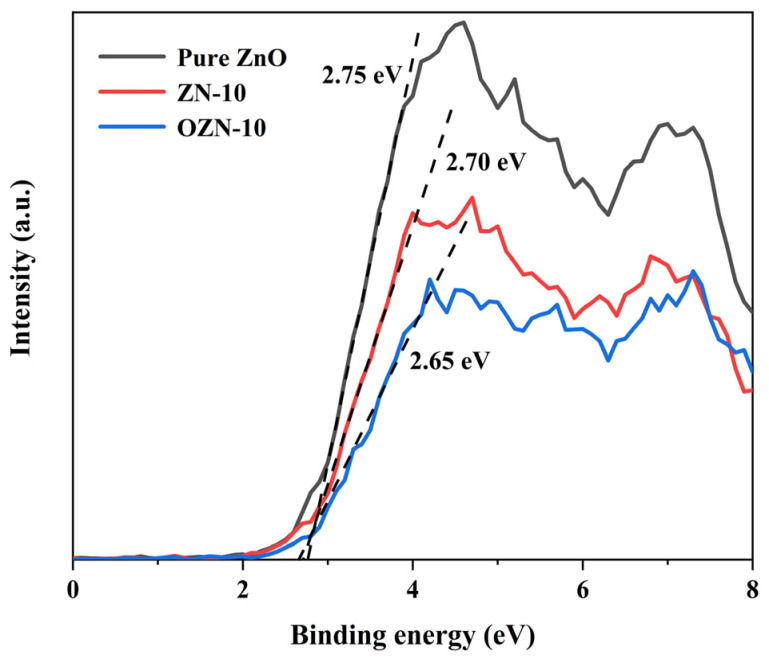
Valence band XPS spectra of pure ZnO, ZN-10 and OZN-10.

**Figure 13 nanomaterials-12-00433-f013:**
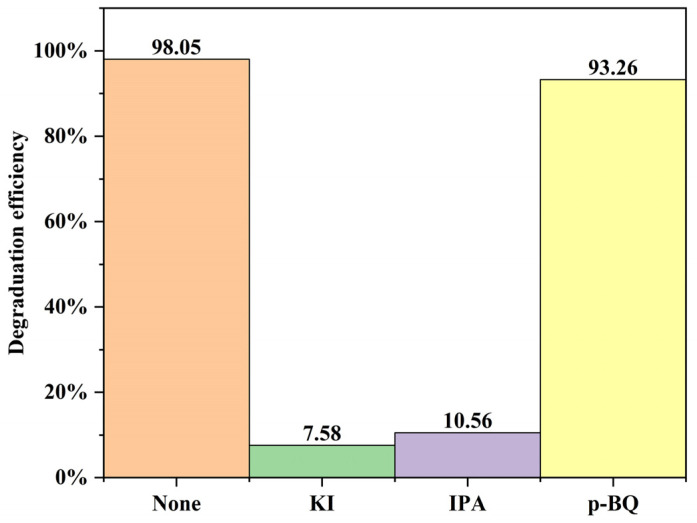
Degradation rate constants of different free radical trapping agents on OZN-10.

**Figure 14 nanomaterials-12-00433-f014:**
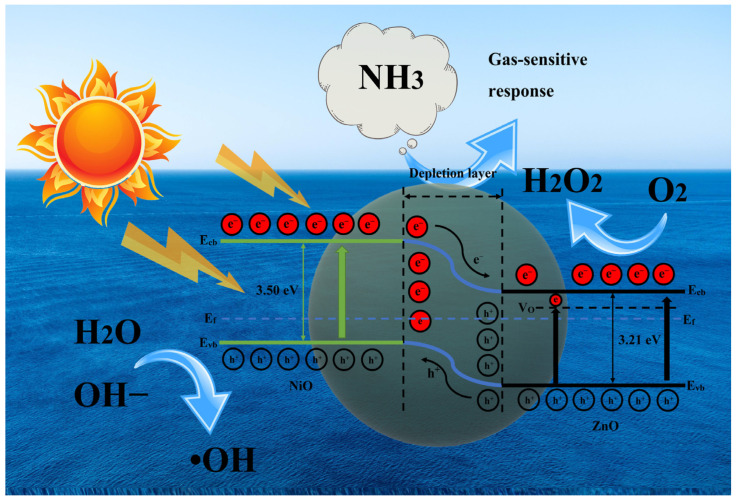
The mechanism diagram of ZnO/NiO shows the photocatalytic mechanism and gas sensing, the position of minimum conduction band (E_cb_), maximum valence band (E_vb_), Fermi energy level (E_f_).

**Figure 15 nanomaterials-12-00433-f015:**
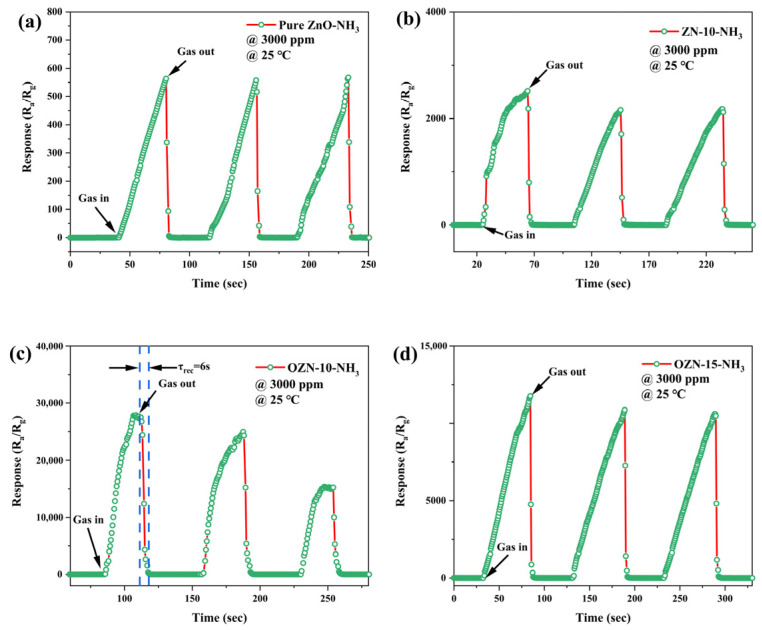
Dynamic response of saturated ammonia gas in (**a**) pure ZnO, (**b**) ZN-10, (**c**) OZN-10 and (**d**) OZN-15 (3000 ppm, 25 °C).

**Figure 16 nanomaterials-12-00433-f016:**
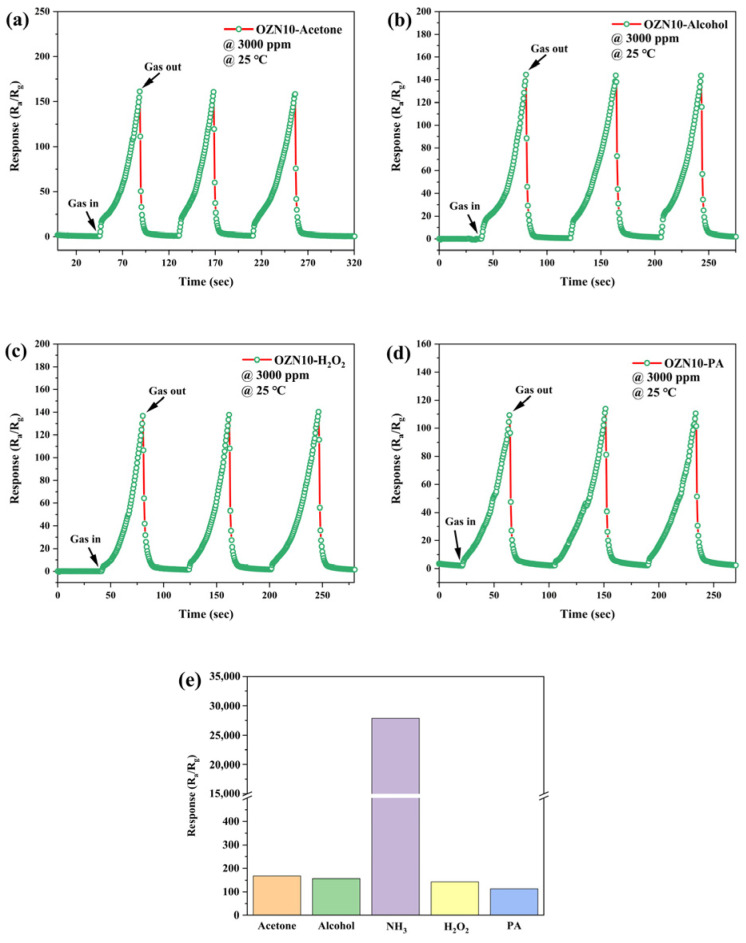
(**e**) Dynamic response diagram and histogram of OZN-10 to saturated (**a**) acetone, (**b**) alcohol, (**c**) H_2_O_2_ and (**d**) PA (3000 ppm, 25 °C).

## Data Availability

No new data were created or analyzed in this study. Data sharing is not applicable to this article.

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
