# Peer review of "The Oxygen Vacancy Defect of ZnO/NiO Nanomaterials Improves Photocatalytic Performance and Ammonia Sensing Performance"

_nanomaterials, 2022, doi:10.3390/nano12030433_

Round 1

Reviewer 1 Report

In the current work, Zhang and Li prepared oxygen vacancy defect of ZnO/NiO nanomaterials composite that improves the photocatalytic performance and ammonia sensing performance. The organization of the paper is good, and the results are clearly presented. I would recommend publication in its current form.

Author Response

Thank you very much for taking time out of your busy schedule to read and evaluate my academic paper. Thank you very much for your recognition and support of my thesis. Your recognition is the biggest encouragement for my scientific research work. We will continue to research and develop semiconductor nanomaterials and further study their mechanism. We also added a few things and made a few minor improvements. Thank you again for your recognition, and I hope I can learn from you in the future.

Reviewer 2 Report

The authors reported ZnO/NiO nanomaterials with oxygen vacancy defect to the photocatalytic performance and ammonia sensing performance. The authors describe that OZN-10  showed  excellent  photocatalytic  performance driven by solar energy and almost completely degraded methylene blue. Their results are seems to be valuable for the scientific community working in this filed. However, i feel that the experimental part of this paper needs more detail. 

in my view, the work is of good worth and may be published.  

Author Response

Thank you very much for taking time out of your busy schedule to read and evaluate my academic paper. Thank you very much for your recognition and support of my thesis. Your recognition is the biggest encouragement for my scientific research work. We will continue to research and develop semiconductor nanomaterials and further study their mechanisms. In response to your suggestion that some details need to be added in the lab section, we have also added some content and improved some parts. Please refer to the attachment.

Reviewer 3 Report

  1. In the abstract, please mention the photocatalytic efficiency and sensing parameters (response value, ppm, temperature, response-recovery times) for the best sample.
  2. The advantages of metal oxide p-n heterojunction in gas sensing should be explained properly in introduction section using suitable recent references e.g. https://doi.org/10.1016/j.colsurfa.2020.125962 https://doi.org/10.1016/j.snb.2021.130264
  3. How about performance of pure NiO for photocatalysis and gas sensing as compared to ZnO/NiO?
  4. 4 c), please identify the SAED diffraction rings assigned to those hkl planes.
  5. Why gas response are so spikey in nature and triangular in nature? Why there is not plataue of saturation?
  6. Why recovery is so fast?
  7. How oxygen defects can be quantitatively analyzed? EDS atomic percentage etc?
  8. Where is Fig. 15?
  9. Caption of Fig. 16 should be in order and more clear at what temperature? Which concentrations etc?
  10. The very important characterization is XRD which determines crystal structure as well as various other parameters e.g. average crystallite size, preferred growth, micro strain. XRD analysis should be elaborated. As given in https://doi.org/10.1016/j.nanoso.2018.01.007
  11. The amount of sensor material loaded should be mentioned with thickness of sensing film. This is important in order to understand active area available for sensing.
  12. The sensor resistance of all sensor devices at various temperature should be recorded and plot Resistance Vs Temperature graph.
  13. The details of sensing should be explained. testing volume chamber, which target gases used, commercially available or lab prepared, how ppm level maintained and injected, resistance measured by which system with what time interval schematic and real picture of testing set-up will be helpful to understand better.
  14. The original photographic pictures of all sensor devices.
  15. Finally, some grammatical, typo mistakes should be resolved and scientific English should be re-checked.

Author Response

Thank you very much for taking time out of your busy schedule to read and evaluate my academic paper. Thank you very much for your recognition and support of my thesis. Your recognition is the biggest encouragement for my scientific research work. We will also continue to research and develop semiconductor nanomaterials and further study their mechanisms. In view of your valuable suggestions and questions, we have also made supplements and modifications to the relevant content, and improved some of the content. Thank you again for your professional suggestions. All of your suggestions are very important. They have important guiding significance for my future scientific research work and enrich my paper. Please see the attachment.

Round 2

Reviewer 3 Report

The manuscript nanomaterials-1562131 entitled "The oxygen vacancy defect of ZnO/NiO nanomaterials improves the photocatalytic performance and ammonia sensing performance” is well revised. A significant amount of study has been carried out. The authors have addressed all concerns. Hence, I recommend the present revised version of the manuscript for publication in the Nanomaterials. 

Recommendation: Accept.